# Uncovering Prototypical Knowledge for Weakly Open-Vocabulary Semantic Segmentation

**Fei Zhang**[1]    **Tianfei Zhou**[3]    **Boyang Li**[4]    **Hao He**[5]    **Chaofan Ma**[1]    **Tianjiao Zhang**[1]
**Jiangchao Yao**[1,2†]    **Ya Zhang**[1,2]    **Yanfeng Wang**[1,2]

[1]CMIC, Shanghai Jiao Tong University    [2]Shanghai AI Laboratory
[3]Beijing Institute of Technology    [4]National University of Defense Technology    [5]CUHK

ferenas@sjtu.edu.cn,  ztfei.debug@gmail.com
{chaofanma, xiaoeyuztj, Sunarker, ya_zhang, wangyanfeng622@}@sjtu.edu.cn

## Abstract

This paper studies the problem of *weakly open-vocabulary semantic segmentation* (WOVSS), which learns to segment objects of arbitrary classes using mere image-text pairs. Existing works turn to enhance the vanilla vision transformer by introducing explicit grouping recognition, i.e., employing several group tokens/centroids to cluster the image tokens and perform the group-text alignment. Nevertheless, these methods suffer from a *granularity inconsistency* regarding the usage of group tokens, which are aligned in the all-to-one *v.s.* one-to-one manners during the training and inference phases, respectively. We argue that this discrepancy arises from the lack of elaborate supervision for each group token. To bridge this granularity gap, this paper explores explicit supervision for the group tokens from the *prototypical knowledge*. To this end, this paper proposes the *non-learnable prototypical regularization* (NPR) where non-learnable prototypes are estimated from source features to serve as supervision and enable contrastive matching of the group tokens. This regularization encourages the group tokens to segment objects with less redundancy and capture more comprehensive semantic regions, leading to increased *compactness* and *richness*. Based on NPR, we propose the *prototypical guidance segmentation network* (PGSeg) that incorporates multi-modal regularization by leveraging prototypical sources from both images and texts at different levels, progressively enhancing the segmentation capability with diverse prototypical patterns. Experimental results show that our proposed method achieves state-of-the-art performance on several benchmark datasets. The source code is available at https://github.com/Ferenas/PGSeg.

## 1 Introduction

Recently, the remarkable success of *vision-language pre-training* (VLP) methods [39, 29, 2] has invigorated the field of semantic segmentation, one of the prominent computer vision tasks. This advancement has led to the emergence of an intriguing task known as *open-vocabulary semantic segmentation* (OVSS), which aims to segment object pixels belonging to arbitrary classes beyond pre-defined categories. To address this challenge, most works [22, 19, 34, 37] have turned to a large quantity of image-text pairs equipped with precisely-annotated masks. To liberate OVSS from exhaustive pixel-level ground truth, we in this paper excavate *weakly open-vocabulary semantic segmentation* (WOVSS), a more arduous setting that achieves OVSS with mere image-text pairs.

---

† denotes the corresponding author

37th Conference on Neural Information Processing Systems (NeurIPS 2023).

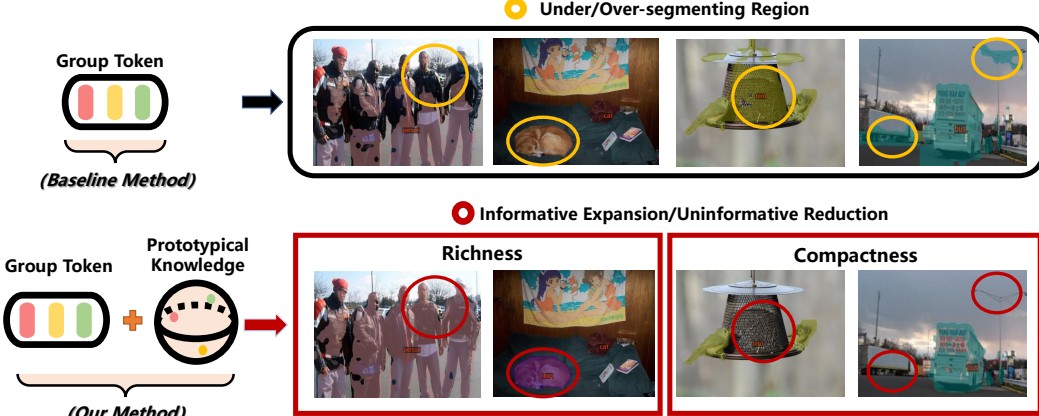

Figure 1: Illustration of our motivation. Our method explores the *prototypical knowledge* to provide explicit supervision for the group tokens, which improves the segmentation results with the increased *richness* and *compactness*. The former improves the feature representation of group tokens, resulting in enlarged semantic regions, and the latter reduces cluster redundancy and noise.

To learn from vast image-text data, *vision transformer* (ViT) [14] has exhibited impressive advances in acquiring powerful visual representations from the text [39, 29]. However, the vanilla ViT lacks an explicit grouping component, rendering it inadequate for achieving comparable fine-grained segmentation with only text supervision. To imbue ViT with the potential for segmenting ability, most WOVSS approaches [33, 50, 41, 51, 35] proposed to cluster the patch-level visual features into several learnable group tokens/centroids, and process the group-text alignment to generate the corresponding categories. Though effective, these methods inevitably are plagued with a *granularity inconsistency* concerning the group tokens. During the training stage, these learnable group tokens are averaged to facilitate all-to-one group-text alignment, while a one-to-one alignment strategy is employed during zero-shot inference (please refer to Figure 2 for more details). This inconsistency arises due to the weak supervision inherent in WOVSS, otherwise, they could be regularized with, e.g. pixel-level ground truth, to perform promising segmentation results as normal OVSS methods do [9, 28]. To break this ill-regularized learning pattern, this paper aims to bridge the granularity gap by exploring explicit supervision for the group tokens, remedying the flawed supervision in WOVSS.

Before delving into the proper guidance for group tokens, let us revisit an interesting question: *What constitutes a good cluster?* Such a question drives us to put forward two properties that a reliable group centroid should possess. 1) *Compactness* ensures that a cluster centroid and its clustered items are closely located in the feature space, forming a compact cluster with minimal noise and redundancy [31, 24, 30]. 2) *Richness* refers to the capability of a centroid to capture diverse and accurate patterns, thereby enhancing zero-shot generalization capability [3, 4, 56]. These two properties motivate us to find the supervision by exploiting the *prototypical knowledge* [4, 31, 30] from an *expectation-maximization* (EM) estimated data density. To this end, we propose the *non-learnable prototypical regularization* (NPR) that adopts the *Gaussian mixture models* (GMM) [42], one of the soft clustering models, to generate the supervision from the source features for each group token. Specifically, we treat the learned Gaussian distributions from the source features as prototypes, which are then used to align with the group tokens in a contrastive manner. Notably, each non-learnable prototype (uninvolved in gradient backpropagation) is able to regularize the corresponding group token, enabling it to segment compactly and richly. As shown in Figure 1, the group tokens could benefit from this prototypical supervision to segment the objects with less redundancy and more accurate semantic patterns, effectively alleviating the under- and over-segmentation problems.

To instantiate the prototypical patterns in NPR, this paper introduces a novel investigation into using multi-modal information as the source features. The low-level image features, with detailed texture information, could be an intuitive choice to refine the segmentation results [44, 49, 1]. Beyond such simple uni-modality, we further mine out the prototypes from the text to regularize the group token with textual information. Hence we propose two strategies, namely *image-level* NPR (I-NPR) and *text-level* NPR (T-NPR), to provide multi-modality regularization to the group tokens. Based on this, we propose the *prototypical guidance segmentation network* (PGSeg), a hierarchical segmentation model that incorporates I-NPR and T-NPR into the group tokens at different levels, progressively improving the segmenting ability of the group tokens. Overall, we make the following contributions:

- We propose NPR that explores and exploits the *prototypical knowledge* to serve as a valid supervision of the group tokens in segmenting objects. This explicit regularization is encouraged to bring compact and rich feature representation to the group tokens.

- We propose PGSeg, a simple yet effective segmentation architecture that extracts the *prototypical knowledge* from both the image and text to regularize the group tokens at different levels, progressively guiding the group tokens to segment in an explicit way.

- Extensive results on several benchmarks demonstrate the superiority and effectiveness of our method. Particularly, our method yields new state-of-the-art performance by reaching **53.2**% mIoU and **28.7**% mIoU on PASCAL VOC12 [16] and COCO [32], respectively. It is worth highlighting that our PGSeg model, trained solely on the CC12M dataset [7], surpasses some state-of-the-art methods utilizing large foundation models such as CLIP [39] and BERT [13] by up to **14.5%** and **5.2%** in terms of mIoU on PASCAL VOC12 and COCO, respectively.

## 2 Related Work

**Weakly Open-Vocabulary Semantic Segmentation.** Most existing works addressing WOVSS can be categorized into two groups based on whether CLIP [39] or Diffusion Models [43] is employed as the fundamental model. The first category focuses on extracting coarse localization features from CLIP or Stable Diffusion Models and subsequently refining them to achieve fine-grained segmentation results [55, 6, 36]. The second category of approaches, distinct from those focused on CLIP, centers around enhancing the plain ViT by incorporating grouping recognition, resulting in a foundational segmentation model [33, 50, 41, 51]. In these methods, several learnable group tokens/centroids are introduced to extract visual concepts from the image features. [50] proposed GroupViT that assigns these tokens to input patch tokens, enabling a learnable clustering process during training. [35] also presented a grouping-based approach, and introduced a reconstruction loss and a superpixel-based regularization loss to improve the inner clustering results. Our work aligns with the second category of approaches. Note that the setting of WOVSS is extremely similar to *weakly supervised semantic segmentation* (WSSS), where a segmentation model is obtained with simply image-level labels. Most works addressing WSSS aim to use the low-level of the image information to iteratively refine the segmentation mask [44, 1, 54], requiring massive additional training or inference stages on the target dataset. Therefore, this paper explores an end-to-end mechanism that efficiently incorporates low-level information on the segmentation mask.

**Prototypes for Deep Representation Learning.** The prototypes typically refer to the centroids from conventional clustering methods [15]. Based on the *expectation-maximization* (EM) algorithm, prototypes are learned by estimating the data features through a mixture of prior distributions. As a result, prototypes are often regarded as "non-learnable" since they deviate from the usual gradient-based learning in deep neural networks [31, 56]. The inclusion of prototypical patterns has been extensively explored to enhance feature representation learning in *contrastive learning* (CL) [3, 4, 5, 31, 57], which aims to match the feature embeddings of a pair of aligned samples. The success of these approaches highlights two important benefits that prototypes bring to feature alignment. The first goes to the *compactness* [31, 30, 24], where they find that the prototypes could reformulate features into a more compact representation, reducing redundancy and noise in feature alignment. This leads to more reliable feature representations. Another benefit is to enhance the *richness* of the feature representation by capturing more learning patterns. CL often suffers from the *dimensional collapse*, where embedding vectors occupy a lower-dimensional subspace than their original dimension, resulting in limited diversity in feature representation. To address it, a line of work has leveraged the prototypes to serve as a constraint on the feature alignment, which is validated to effectively enrich the feature representation [3, 4, 5, 53]. This work explores the use of *prototypical knowledge* with the expectation of providing the aforementioned advantages to segmentation clusters.

## 3 Rethinking the Semantic Grouping Mechanism in WOVSS

To effectively tackle WOVSS, recent works [50, 41, 35, 51] have placed significant emphasis on incorporating explicit grouping recognition into the plain model. To this end, these methods developed a *semantic grouping mechanism* (SGM) based on ViT [14]. Formally, given $m$ input patch tokens $\boldsymbol{S} = [\boldsymbol{s}_1, \boldsymbol{s}_2, .., \boldsymbol{s}_m] \in \mathbb{R}^{m \times d}$ and extra $q$ learnable group tokens $\boldsymbol{G} =$

$[\boldsymbol{g}_1, \boldsymbol{g}_2, ..., \boldsymbol{g}_q] \in \mathbb{R}^{q \times d}$, where $d$ is the dimension of data and $q < m$. SGM clusters $\boldsymbol{S}$ and outputs new clustered tokens $\hat{\boldsymbol{S}} \in \mathbb{R}^{q \times d}$ by $\hat{\boldsymbol{S}} = \texttt{Gumbel-Softmax}(\mathcal{Q}(\boldsymbol{G})\mathcal{K}(\boldsymbol{S})^{\top})\mathcal{V}(\boldsymbol{S}) + \boldsymbol{G}$, where $\mathcal{Q} : \mathbb{R}^{q \times d} \to \mathbb{R}^{q \times d}, \mathcal{K} \, (\mathcal{V}) : \mathbb{R}^{m \times d} \to \mathbb{R}^{m \times d}$ represent the *Query*, *Key* (*Value*) mapping function. Figure 2 clearly demonstrates this cross-attention-based clustering process. Here each patch token is assigned to a corresponding group token by the Straight-Through `Gumbel-Softmax` [23], making this process end-to-end trainable. We formulate this patch-group assignment as $\boldsymbol{A} = \mathcal{Q}(\boldsymbol{G})\mathcal{K}(\boldsymbol{S})^{\top} \in \mathbb{R}^{q \times m}$. By plugging the SGM into ViT, the vanilla image encoder could be empowered with potential segmenting ability. However, it could be observed that this mechanism presents a *granularity inconsistency* for the group tokens between the training and inference stages (as shown in Figure 2). More specifically, during the training phase, all group tokens are globally averaged to match the corresponding text embedding for the final group-text alignment,

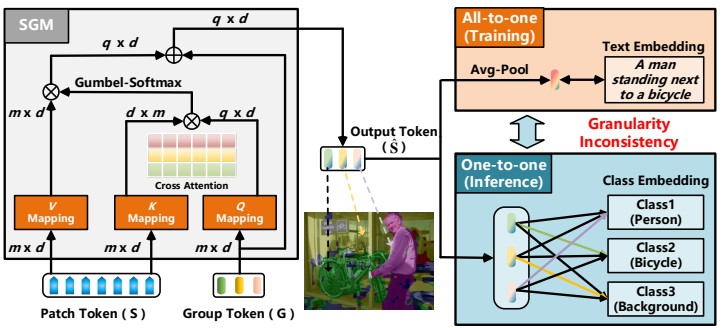

while during the inference phase, each group token necessitates a separate comparison with each class embedding in order to acquire the semantic label, which is then used to annotate the corresponding image regions based on the patch-group assignment. Consequently, the group tokens used in the one-to-one group-text alignment do not receive explicit supervision

Figure 2: The *granularity inconsistency* of SGM.

as they are subjected to all-to-one text-based regularization during the training stage. This discrepancy in supervision may contribute to the performance gap observed between OVSS and WOVSS. In OVSS, each learnable group token can be treated as a query embedding, generating a dense embedding for object mask prediction. This dense embedding can be further regularized by the patterns extracted from pixel-level ground truth annotations [9, 28, 58]. Therefore, such granularity discrepancy essentially is derived from the weak supervision of WOVSS. Despite the challenges posed, we are motivated to explore customized regularization techniques for each group token, aiming to compensate for the absence of pixel-level annotations. By explicitly addressing the granularity gap, we aim to enhance the segmentation performance in WOVSS.

# 4 Methods

## 4.1 Exploring Prototypical Knowledge for Explicit Supervision

To find explicit and reliable supervision, we turn to the *prototypical knowledge* to form a regularized basis that can bring certain benefits to the group token in segmentation. *Gaussian mixture models* (GMM) [42] has been experimentally validated to form a basis that could reduce the redundancy of the features [31, 24]. Inspired by this, we propose the *non-learnable prototypical regularization* (NPR) that uses GMM to extract the prototypes from the prototypical source (like a way of data mining), and then aligns such prototypes with the group centroids in a contrastive way.

**Prototype Generation.** The first stage of NPR is to generate the supervision by GMM. GMM is based on a mixture of Gaussian distributions, where each component of the mixture corresponds to a cluster in the data. Formally, given the prototypical source features $\boldsymbol{V} = [\boldsymbol{v}_1, ..., \boldsymbol{v}_m] \in \mathbb{R}^{m \times d}$, and extra $q$ randomly-initialized prototypes $\boldsymbol{P} = [\boldsymbol{p}_1, ..., \boldsymbol{p}_q] \in \mathbb{R}^{q \times d}$, where $m$ and $d$ represents the number and the dimension of the prototypical source. In this way, the distribution of $\boldsymbol{V}$ could be expressed as $p(\boldsymbol{V}) = \sum_{i=1}^{q} \pi_i \mathcal{N}(\boldsymbol{V}|\boldsymbol{p}_i, \boldsymbol{\Sigma}_i)$, where $\pi_i \in \mathbb{R}^1$, $\boldsymbol{p}_i \in \mathbb{R}^d$ and the $\boldsymbol{\Sigma}_i \in \mathbb{R}^{d \times d}$ are the weight, mean and covariance of the $i$-th Gaussian component. Here we take the means as the prototypes. To work out $(\boldsymbol{P}, \boldsymbol{\Sigma}, \pi)$, the log-likelihood of $p(\boldsymbol{V})$ is maximized through the *expectation-maximization* (EM) algorithm, which optimizes the model, until convergency is reached, by alternating between an *expectation* (E) step and a *maximization* (M) step. In the E step, the probability of $j$-th source feature

---

**Algorithm 1** Non-learnable Prototypical Regularization (NPR)

---

**Require:** Group tokens $\boldsymbol{G} \in \mathbb{R}^{q \times d}$, prototypes $\boldsymbol{P} \in \mathbb{R}^{q \times d}$, prototypical source features $\boldsymbol{V} \in \mathbb{R}^{m \times d}$, iterations $T$ ($T = 10$ in our setting), selecting threshold $\phi$.
1: ▷ **Prototype Generation**
2: **for** iteration $t = 1$ to $T$ **do**
3:      **(E-step) Calculate** the probability of $\boldsymbol{V}$ belonging to $\boldsymbol{P}$ in Eq. (1)
4:      **(M-step) Update** the prototypes $\boldsymbol{P}$ by using Eq. (2)
5: **end for**
6: ▷ **Prototype Supervision**
7: **Generate** the matched prototypes $\boldsymbol{P}^{\mathrm{h}}$ by using the Hungarian matching between $\boldsymbol{P}$ and $\boldsymbol{G}$
8: **Select** the matched pairs ($\boldsymbol{P}^{\mathrm{h}}$, $\boldsymbol{G}$) whose similarity scores are beyond $\phi$
9: **Regularize** the selected $\boldsymbol{G}$ with the matched $\boldsymbol{P}^{\mathrm{h}}$ by using $\mathcal{L}_{\mathrm{PG}}$ in Eq. (3)

---

$\boldsymbol{v}_j$ belonging to the $i$-th Gaussian prototype could be expressed as

$$y_{ij} = \frac{\mathcal{Z}(\boldsymbol{v}_j | \boldsymbol{p}_i)}{\sum_{i=1}^{q} \mathcal{Z}(\boldsymbol{v}_j | \boldsymbol{p}_i)} = \frac{\exp(\boldsymbol{p}_i \boldsymbol{v}_j^{\top})}{\sum_{i=1}^{q} \exp(\boldsymbol{p}_i \boldsymbol{v}_j^{\top})}, i \in \{0, ..., q\}, j \in \{0, ..., m\}, \tag{1}$$

where $\mathcal{Z} : \mathbb{R}^{1 \times d} \to \mathbb{R}^1$ denotes a kernel function. For simplicity, we set $\boldsymbol{\Sigma}$ as the identity matrix $\boldsymbol{I}$ and leave out $\pi$. We experimentally observe that the different choices of $\mathcal{Z}$ negligible effects on the final results, so we simplify the Gaussian Kernel $\exp\left(-\|\boldsymbol{x} - \boldsymbol{y}\|^2 / 2\sigma^2\right)$ to the exponential inner dot $\exp\left(\boldsymbol{x}\boldsymbol{y}^{\top}\right)$. Based on the estimated $y_{ij}$, the prototypes $\boldsymbol{P}$ in the M step could be updated as

$$\boldsymbol{p}_i = \frac{\sum_{j=1}^{m} y_{ij} \boldsymbol{v}_j}{\sum_{j=1}^{m} y_{ij}}. \tag{2}$$

After alternatively repeating the E step and M step, a bunch of compact prototypes representing the prototypical information could be obtained to supervise the group tokens. Note that we could reconstruct the prototypical source $\boldsymbol{V}$ by $\boldsymbol{V}' = \boldsymbol{Y}^{\top} \boldsymbol{P} \in \mathbb{R}^{m \times d}, \boldsymbol{Y} = [y_{ij}]_{q \times m}$.

**Prototype Supervision.** The second stage is to regularize $\boldsymbol{G}$ with the updated $\boldsymbol{P}$. Based on the cosine similarity between $\boldsymbol{P}$ and $\boldsymbol{G}$, we first perform Hungarian matching [27] to ensure that each centroid $\boldsymbol{g}$ has a corresponding target prototype $\boldsymbol{p}$. Denote the matched prototypes as $\boldsymbol{P}^{\mathrm{h}} = [\boldsymbol{p}_1^{\mathrm{h}}, ..., \boldsymbol{p}_q^{\mathrm{h}}] \in \mathbb{R}^{q \times d}$. Then, we combine the matched pair of $(\boldsymbol{g}, \boldsymbol{p}^{\mathrm{h}})$ as the positive samples, and propose the *Prototypical Guidance* (PG) loss $\mathcal{L}_{\mathrm{PG}}$ to regularize the group centroids in a contrastive way:

$$\mathcal{L}_{\mathrm{PG}}(\boldsymbol{G}, \boldsymbol{P}^{\mathrm{h}}) = -\frac{1}{q} \sum_{i=1}^{q} (\log \frac{\exp(\mathcal{S}(\boldsymbol{g}_i, \boldsymbol{p}_i^{\mathrm{h}})/\tau)}{\sum_{j=1}^{q} \exp(\mathcal{S}(\boldsymbol{g}_i, \boldsymbol{p}_j^{\mathrm{h}})/\tau)} + \log \frac{\exp(\mathcal{S}(\boldsymbol{p}_i^{\mathrm{h}}, \boldsymbol{g}_i)/\tau)}{\sum_{j=1}^{q} \exp(\mathcal{S}(\boldsymbol{p}_i^{\mathrm{h}}, \boldsymbol{g}_j)/\tau)}), \tag{3}$$

where $\tau = 0.1$ is the temperature hyper-parameter, and $\mathcal{S}(\boldsymbol{a}, \boldsymbol{b}) = \frac{\boldsymbol{a}\boldsymbol{b}^{\top}}{\|\boldsymbol{a}\|\|\boldsymbol{b}\|}$ calculates the cosine similarity between two vectors. Based on the PG loss, we introduce a simple *hard rejecting strategy* (HRS) that only considers the positive pairs whose similarity is beyond a fixed threshold $\phi$. We claim that one group centroid could be wrongly guided by the matched prototype once they have a significant difference, which will be discussed in Section 5.3. Besides, we here assume that the number of prototypes and group centroids is the same, and we will also show the case where the number of them is different in Section 5.3 (only the matched pairs are considered to calculate $\mathcal{L}_{\mathrm{PG}}$).

**Compactness & Richness.** The essence of NPR is to regulate each group centroid with a normalized prototype from a prior distribution, yielding two essential benefits as discussed 2. The first one is the *compactness*, which helps refine the clustered results by reducing noise and redundancy [31, 24]. The second one goes to the *richness* that empowers the group tokens with rich feature representation by relieving the *dimensional collapse* through the application of normalized regularization [4, 18], capturing more accurate patterns as possible. In all, we believe that NPR could augment the segmenting ability of the group tokens with these two benefits, which will be validated in Section 5.

**Complexity Analysis.** The EM algorithm is the key part in NRP (Prototype Generation in Algorithm 1). It is crucial to carefully consider its time complexity regarding iterative learning. However, through a simplified implementation, we demonstrate that the time complexity of prototype generation in NPR is $\mathcal{O}(q \times m \times d)$ for a single sample. This complexity is reasonably acceptable for implementation purposes. The actual computational performance will be demonstrated in Section 5.3.

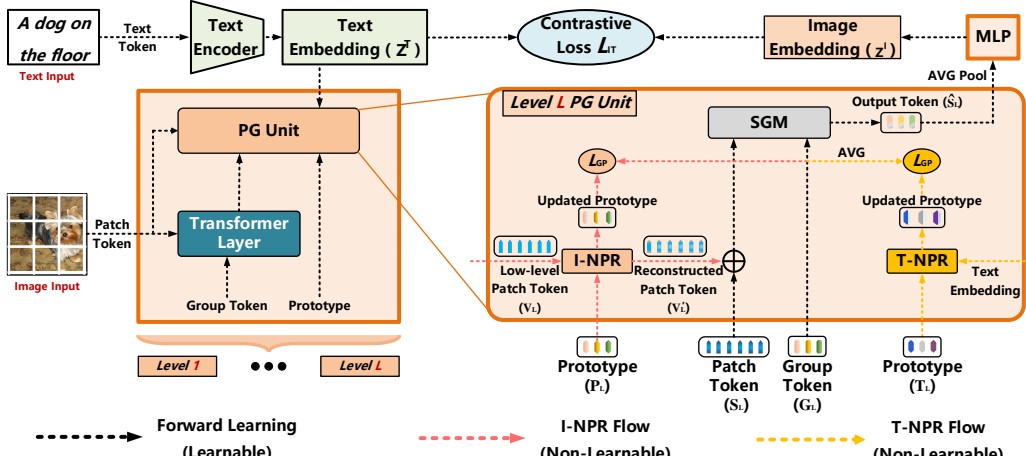

Figure 3: The overall framework of our proposed PGSeg. PGSeg consists of an image encoder and a text encoder. Through sequential connections facilitated by the PG Unit and several transformer layers, the image encoder is organized into multiple hierarchical levels (highlighted in orange), enabling the progressive grouping of the input patch tokens. Best viewed in color.

## 4.2 Exploiting Prototypical Guidance for WOVSS

In this section, we introduce our proposed *prototypical guidance segmentation network* (PGSeg) in detail, which incorporates the proposed NPR into SGM to address WOVSS.

**Main Architecture.** As shown in Figure 3, the overall framework of PGSeg is mainly comprised of a text encoder and an image encoder. The transformer-based text encoder is used to output the text embedding, denoted as $\boldsymbol{Z}^{\mathrm{t}} \in \mathbb{R}^{n \times c}$, where $n$ denotes the sample batch size and $c$ denotes the feature dimension. For the image encoder, we adopt a ViT-based architecture as the image encoder. To instill the image encoder with the segmenting ability, we propose PG Unit, which is a plug-and-play module to perform grouping based on SGM. Intuitively, a multitude of PG Units, along with several transformer layers, can be sequentially connected to perform hierarchical grouping during the forward learning process. The image embedding $\boldsymbol{Z}^{\mathrm{i}} \in \mathbb{R}^{n \times c}$, as the output of the image encoder, is generated by average-pooling and mapping the output tokens in the final PG Unit. Based on the architecture of PGSeg, we assume that the image encoder could be split into $L$ levels if $L$ PG Units are inserted. Formally, we denote the input tokens in the $l$-th level as $\mathbf{S}_l \in \mathbb{R}^{n \times m_l \times d}$, and $q_l$ learnable group tokens as $\mathbf{G}_l \in \mathbb{R}^{n \times q_l \times d}$, where $m_l$ ($q_l$) denotes the number of input patch (group) tokens in level $l$. Likewise, denote the output tokens in the $l$-th level as $\hat{\mathbf{S}}_l \in \mathbb{R}^{n \times q_l \times d}$. Intuitively, we hold that $\mathbf{S}_{l+1} = \hat{\mathbf{S}}_l, l \in \{1, ..., L\}$ due to the sequential connection among the PG Units, namely the output tokens in the previous level serve as the input patch tokens in the next level.

**PG Unit.** To instantiate NPR, we propose PG Unit that mines out the multi-modal prototypical source from image-text representations. In this way, we propose two NPR-based strategies based on the type of prototypical pattern, namely *image-level NPR* (I-NPR) and *text-level NPR* (T-NPR). For I-NPR, we adopt the input tokens placed before the transformer layers as the image-level prototypical pattern, which are denoted as $\mathbf{V}_l \in \mathbb{R}^{n \times m_l \times d}$. Based on extra non-learnable prototypes $\mathbf{P}_l \in \mathbb{R}^{n \times q_l \times d}$, we follow the Algorithm 1 and regularize $\mathbf{G}_l$ with $\mathbf{V}_l$ and $\mathbf{P}_l$. Besides, we further reform the input tokens by $\mathbf{S}_l = \mathbf{S}_l + \mathbf{V}'_l$, where $\mathbf{V}'_l \in \mathbb{R}^{n \times m_l \times d}$ are the reconstructed image-level source, as the input tokens of SGM to enhance the robustness of the model learning [10].

For T-NPR, we turn to the text embedding $\boldsymbol{Z}^{\mathrm{t}} \in \mathbb{R}^{n \times c \times 1}$ as the text-level prototypical pattern to improve the group tokens $\mathbf{G}_l$ in capturing semantic information. Specifically, we introduce additional text prototypes $\mathbf{T}_l \in \mathbb{R}^{n \times q_l \times 1}$ and update them with $\boldsymbol{Z}^t$. Subsequently, we regularize $\mathtt{AVG}(\mathbf{G}_l)$ with the updated $\mathbf{T}_l$, where $\mathtt{AVG} : \mathbb{R}^{n \times q_l \times d} \to \mathbb{R}^{n \times q_l \times 1}$ averages the $\mathbf{G}_l$ along dimension $d$. Essentially, T-NPR aligns the score of each group token with the center value clustered from dimension $d$ in $\mathbf{Z}^t$.

**Training Loss.** Based on the proposed I-NPR and T-NPR in PG Unit, the overall training loss is

$$\mathcal{L}_{\mathrm{ALL}} = \mathcal{L}_{\mathrm{IT}}(\boldsymbol{Z}^{\mathrm{t}}, \boldsymbol{Z}^{\mathrm{i}}) + \sum_{l=1}^{L} \sum_{j=1}^{n} \lambda \mathcal{L}_{\mathrm{PG}}(\mathbf{G}_l^j, \mathbf{P}_l^j) + \beta \mathcal{L}_{\mathrm{PG}}(\mathtt{AVG}(\mathbf{G}_l^j), \mathbf{T}_l^j), \qquad (4)$$

where $\mathcal{L}_{\mathrm{IT}}$ is the symmetric image-text contrastive loss in [39], $\lambda$ and $\beta$ are the hyper-parameters to balance the loss. Here we empirically set $\lambda = 0.1$ and $\beta = 0.01$.

**Momentum update of Prototype.** It is critical to set appropriate initialization of $\mathbf{P}$ to guarantee robust convergence [42]. To this end, we leverage the *exponential moving average* (EMA) strategy to globally update the initial prototypes after each training round: $\mathbf{P}_{(\text{new})} = \gamma \mathbf{P}_{(\text{old})} + (1 - \gamma) \sum_{i=1}^{n} \mathbf{P}_{(T)}^{i}$, where $\mathbf{P}_{(T)}^{i}$ denotes the updated prototypes at the final $T$-th iteration in NPR, and $\mathbf{P}_{(\text{new})} / \mathbf{P}_{(\text{old})}$ is the initial prototypes for the next/current training round. Empirically we set $\gamma = 0.9$.

## 5 Experiments

### 5.1 Implementation Details

**Model Architecture.** Following conventions in [50, 41, 35], we adopt ViT-S [47] as the visual backbone, which consists of 12 transformer layers with a hidden dimension of 384. The input image size is 224×224 and the patch size is 16×16. Based on the experimental performance in [50, 41], we set $L = 2$ for PGSeg, which means two individual PG Units are incorporated into the ViT module at different places. The number of learnable group tokens is 64 in level 1 and 8 in level 2, namely $q_1 = 64, q_2 = 8$. Two PG Units are inserted after the $6^{th}$ and $9^{th}$ transformer layers, respectively. The transformer-based text encoder is the same as [39], with the hidden dimension of 256.

**Datasets and Evaluation Metric.** Following [50, 41, 35, 51], we use CC12M [7] and RedCaps [12] as the training sets, and each of them contains 12 million image-text pairs. For the downstream evaluation datasets, we select 5 benchmarks: PASCAL VOC12 (20 foreground classes) [16], PASCAL Context (59 foreground classes) [38], COCO Object (80 foreground classes) [32], ImageNet-S (919 foreground classes) [17] and LVIS [20] (1203 foreground classes). All of them contain 1 extra background class. We use the mean Intersection-over-Union (mIoU) as the evaluation metric.

**Training and Inference Settings.** During the training stage, we use Adam [25] optimizer with a weight decay of 0.05. We set the batch size as 4096, and use the cosine learning strategy with an initial learning rate of $1.6\mathrm{e}^{-3}$. We train the PGSeg for 40 epochs with 5 epochs of linear warm-up. As the generated features are unreliable in early epochs, we set $\lambda = \beta = 0$ at the first 30 epochs. For the selecting threshold $\phi$ of HRS in NPR, we experimentally set it to 0.1. The whole training process is implemented on 4 A100 GPUs, each with 80 GB of memory. During the inference, where no additional training is involved, we obtain the patch-group assignment with the cross-attention maps in two PG Units: $\boldsymbol{A}_{\text{final}} = \boldsymbol{A}_1 \boldsymbol{A}_2 \in \mathbb{R}^{m \times q_2}$. We then generate the foreground semantic mask, and set a fixed threshold as the background score to evaluate the segmentation mask. Note that the templates of the text prompt could significantly impact the segmentation results [40]. To ensure fair comparisons, we follow the same prompt template [50, 39, 41] (*a photo of...*) to generate the text embedding with the foreground class name for the evaluated benchmark datasets.

### 5.2 Zero-shot Segmentation Performance

**Comparison with Zero-shot Baselines.** Similar to [41, 50], we compare our methods with seven baselines. Four of them train a plain ViT and a text encoder simply by $\mathcal{L}_{\text{IT}}$ [39], and generate the segmentation mask by using different pixel-grouping methods on the image features. [52] find that increased [CLS] tokens in ViT serve as meaningful centers for perceptual grouping. Inspired by this, we additionally build a baseline VIT-8S to serve as a more potent parametric baseline. It trains a plain ViT-S but with increasing the [CLS] token amount into 8, and then aligns the averaged embedding of the final 8 [CLS] tokens with the text embedding in a contrastive way. ViT-8S generates the assignment by summing the cross-attention maps ([CLS] to patch) in each transformer layer. Table 1 shows

Table 1: Comparison with zero-shot baselines.

| Architecture | Pixel-Grouping Methods | mIoU (%) |
|---|---|---|
| ViT-S | pixel-wise | 20.1 |
| ViT-S | K-means | 25.0 |
| ViT-S | Mean-shift | 20.7 |
| ViT-S | Spectral clustering | 19.7 |
| ViT-8S | ✗ | 38.1 |
| GroupViT [50] | ✗ | 43.2 |
| ViewCo [41] | ✗ | 45.7 |
| PGSeg | ✗ | **49.0** |

the performance of these methods on the validation set of PASCAL VOC12, note that all methods here are trained simply with CC12M. It is intuitive that PGSeg outperforms all the compared baselines. Notably, PGSeg adopts the same backbone as GroupViT [50] and ViewCo [41], while achieving a **5.8%** and **3.3%** performance improvement over them.

Table 3: Comparison with SOTA in terms of mIoU(%). All the image encoders here are built on ViT-S [47]. *ST* means that [55] uses the *self-training* strategy on the evaluated datasets to refine the mask. The best results are highlighted in **bold** (underline marks the cases under the same volume).

| Methods | Training Data (volume) | Pre-trained Models | VOC12 | Context | COCO |
|---|---|---|---|---|---|
| RECO [45] | CC400M [39] + ImageNet1M (401M) | CLIP [39] + MOCO [21] | 25.1 | 19.9 | 15.7 |
| MaskCLIP [55] | CC400M [39] (400M) | CLIP [39] | 29.3 | 21.1 | 15.5 |
| ViL-Seg [33] | CC12M [7] (12M) | ✘ | 34.4 | 16.3 | 16.4 |
| MaskCLIP [55] | CC400M [39] + *ST* (400M) | CLIP [39] | 38.8 | **23.6** | 20.6 |
| GroupViT [50] | CC12M [7] (12M) | ✘ | 41.1 | 18.2 | 18.4 |
| OVSegmentor [51] | CC12M [7] + ImageNet1M [11] (13M) | BERT [13] + DINO [5] | 44.5 | 18.3 | 19.0 |
| PGSeg | CC12M [7] (12M) | ✘ | 49.0 | 20.6 | **22.9** |
| GroupViT [50] | CC12M [7] + RedCaps12M [12] (24M) | ✘ | 50.8 | 23.6 | 27.5 |
| SegCLIP [35] | CC403M [39, 7] + COCO400k [32] (403.4M) | CLIP [39] | 52.6 | **24.7** | 26.5 |
| GroupViT [50] | CC12M [7] + YFCC14M [46] (26M) | ✘ | 52.3 | 22.4 | 20.9 |
| ViewCO [41] | CC12M [7] + YFCC14M [46] (26M) | ✘ | 52.4 | 23.0 | 23.5 |
| PGSeg | CC12M [7] + RedCaps12M [12] (24M) | ✘ | 53.2 | 23.8 | **28.7** |

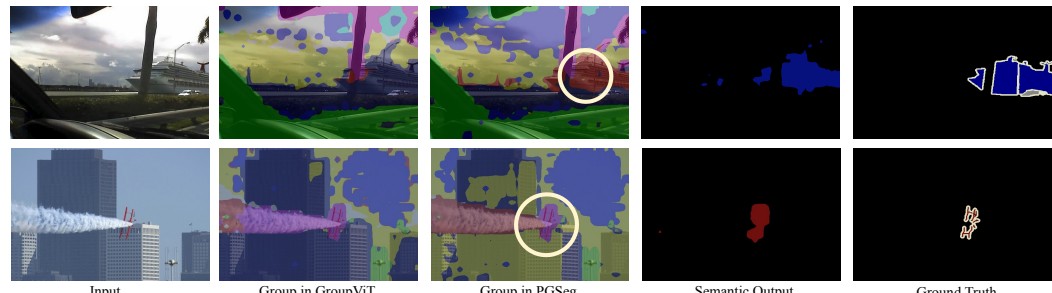

| Input | Group in GroupViT | Group in PGSeg | Semantic Output | Ground Truth |

Figure 4: Qualitative results on PASCAL VOC12. Compared with GroupViT, the group tokens in PGSeg could capture the object (marked with a white circle) in a more complete and delicate way.

**Comparison with SOTA.** Table 3 lists the mIoU of recent state-of-the-art (SOTA) methods on the validation splits of PASCAL VOC12, PASCAL Context, and COCO datasets. Since there is a huge gap among these methods in terms of the datasets and pre-trained model, we particularly report the total data volume and the specific pre-trained model that each method used during the whole training process, with the expectation for a clear and sound fair comparison. As shown in Table 3, our PGSeg achieves the SOTA performance among all the methods with the same data volume. Particularly, our method, with comparably small data volume, even achieves a more stunning performance than the methods with huge foundation models, validating the superiority and effectiveness of our PGSeg.

**Challenging Segmentation.** Here we introduce two challenging benchmarks to explore the potential of PGSeg in real-world segmentation. ImageNet-S, distilled from the ImageNet [11], contains 919 classes of object-centric images with human-annotated mask labels. LVIS contains 1203 classes that are incorporated with various real-world objects. As shown in Table 2, both PGSeg and GroupViT (CC12M + RedCaps12M) are weak in segmenting these two datasets. Such frustrating performance might be due to inadequate image-text alignment of new

Table 2: Challenging segmentation results. TLG refers to the *True Label Guidance* strategy.

| Methods | ImageNet-S (- / +TLG) | LVIS (- / +TLG) |
|---|---|---|
| GroupViT | 19.6 / 32.5 | 3.2 / 6.9 |
| PGSeg | 19.7 / **33.8** | 3.4 / **7.2** |

vocabularies. It is observed that both their segmentation performance could be boosted through the *True Label Guidance*, considering only the true labels of the evaluating samples. It is also found that PGSeg performs better than GroupViT with the TLG strategy. Therefore, we believe that the segmenting performance of PGSeg could be further enhanced by including more image-text pairs.

## 5.3 Ablation Studies

In this section, we use the version of PGSeg with CC12M+RedCaps12M to implement all ablation studies on PASCAL VOC12 in detail, which contain the effectiveness of the modules in the PG Unit, some analysis of the prototypes, and the computational performance of PGSeg.

**Effectiveness of PG Unit.** Table 4 shows the effectiveness of each designed module in the PG Unit. Recall that in Section 4.2 we propose two different NPR-based strategies in the PG Unit, namely I-NPR and T-NPR. As shown in Table 4, both these strategies are effective in enhancing the baseline, delivering **1.75**% and **0.58**% improvements, respectively. We also propose the HRS to further improve the performance of NPR by filtering the group-prototype pairs with a fixed selecting threshold $\phi$. Consequently, it is observed that proper threshold could lead to the boosting of PGSeg (+**0.41**%), which finally achieves **53.24**% performance together with T-NPR.

**The number of Prototypes.** Recall that in Section 4.1 we mention that the number of prototypes could be different from the group tokens. Table 5 reports the performance of PGSeg with different numbers of prototypes. Note that at the $1^{st}$ ($2^{nd}$) level, the number of group tokens remains constant at 64 (8). Here we exclude the HRS to ensure a comprehensive consideration of all prototypes. We only consider the matched group-prototype pairs as the positive samples, and all other extra groups/prototypes would be considered to form the negative samples. In other words, if the number of group tokens is smaller (larger) than the number of prototypes, the symmetric $\mathcal{L}_{PG}$ would simply calculate the left (right) part accordingly. As depicted in Table 5, it has been observed that the optimal performance is attained when the number of prototypes is equal to the number of group tokens. Moreover, any increase or decrease in the number of prototypes beyond this optimal value tends to negatively impact the segmentation performance to some extent. This reveals that the number of negative samples or positive samples is vital to the performance of prototypical alignment. Clearly, the number of positive sample pairs would decrease if the number of prototypes is less than 64/8, otherwise, the number of negative sample pairs would increase. Therefore, our experimental results on the number of prototypes reach a consistent conclusion with [48, 53, 21].

Table 4: Ablation studies on the PG Unit.

| Baseline | I-NPR | T-NPR | HRS-0.5 | HRS-0.3 | HRS-0.1 | mIoU (%) |
|---|---|---|---|---|---|---|
| ✔ | | | | | | 50.76 |
| ✔ | ✔ | | | | | 52.51 |
| ✔ | | ✔ | | | | 51.34 |
| ✔ | ✔ | | ✔ | | | 51.62 |
| ✔ | ✔ | | | ✔ | | 52.57 |
| ✔ | ✔ | | | | ✔ | 52.92 |
| ✔ | ✔ | ✔ | | | ✔ | **53.24** |

Table 5: Ablation studies on the number of prototypes (*w.o.* HRS). The group token amount is 64/8.

| $1^{st}$ level \ $2^{nd}$ level | 4 | **8** | 16 |
|---|---|---|---|
| 32 | 52.12 | 52.49 | 51.94 |
| **64** | 52.71 | **52.83** | 52.23 |
| 128 | 52.21 | 52.62 | 52.14 |

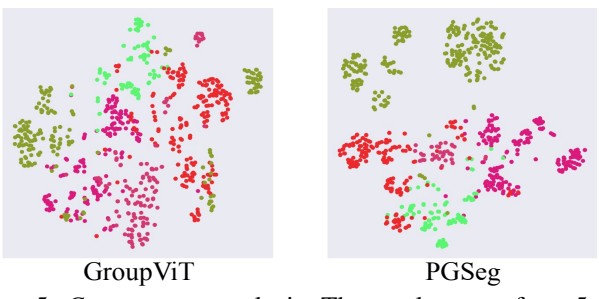

GroupViT  PGSeg

Figure 5: *Compactness* analysis. The results come from 5 clustered patch tokens based on the group tokens.

Figure 6: Dimension distributions of each group token.

**Two benefits of NPR.** Here we aim to investigate the nature of the two benefits, i.e., *compactness* and *richness* (mentioned in Section 4.1), to understand the effectiveness of NPR better. To validate the first one, we use tSNE to visualize the clustered 5 input patch tokens of SGM among 50 samples. As shown in Figure 5, it is intuitively found that the patch tokens in PGSeg are more tightly clustered than GroupViT, where most input tokens are comparably scattered. With the help of the compact basis in NPR, the input patch tokens become less noisy in the feature space, which is also supported by the visualized results in Figure 4 and Figure 1. The second benefit aims to enrich the feature representation to capture more accurate semantic patterns by relieving the *dimensional collapse* [8, 18]. As shown in Figure 6, we calculate the mean and variance of 384 dimensions for each $1^{st}$ level group token (64 in total) based on 300 samples. It is evident that although the means are nearly identical, the dimensional variance of PGSeg is significantly larger than that of GroupViT, indicating a better dimensional representation for each group token with prototypical regularization. Overall, these two explicit benefits are validated to contribute to the enhanced segmenting ability of the group tokens.

**Computational Performance.** Here we present the floating-point operations (FLOPs) and model parameters for three methods in Table 6 as well. The FLOPs are calculated based on an image size of $448 \times 448$. It is observed that PGSeg introduces a 3.2G increase in FLOPs compared to GroupViT, but still maintains a lower FLOP count (-3.1G) than ViT-S. Therefore, the computational complexity of PGSeg remains manageable, indicating a reasonable level of computational efficiency.

Table 6: Computational performance.

| Methods | FLOPs | Params |
|---------|-------|--------|
| ViT-S | 16.7G | 21.6M |
| GroupViT | 10.4G | 28.7M |
| PGSeg | 13.6G | 35.1M |

## 6 Discussion with SAM

Recently, the Segment Anything Model (SAM) [26], an impressive model for interactive segmentation, has demonstrated significant progress in image segmentation tasks. SAM supports segmenting everything in an arbitrary image, which is a powerful foundation model to address OVSS. SAM is trained on a massive dataset comprising **11 billion** images. In contrast, PGSeg is trained on a smaller dataset consisting of only **24 million** image-text pairs. Intuitively, the data volume of SAM is approximately **460 times** that of PGSeg. Despite their vast data amount, SAM also incorporates over 1 billion accurately annotated masks. Therefore, SAM is clearly better than PGSeg. Though a huge performance gap between our PGSeg and SAM, we would like to present some comparisons between these two models to present the research value of WOVSS. As illustrated in Figure 7, the segmenting groups in SAM provide comprehensive coverage of objects in an extremely fine-grained manner. In comparison, our learnable groups effectively capture entire objects without requiring instance-level recognition. For example, in the $1^{st}$ column of the image, our yellow group can represent the overall forest background, while SAM can differentiate between individual trees within the forest background. However, it is important to note that our PGSeg model achieves comparable segmentation capabilities for certain intuitive objects, such as umbrellas, ships, babies, etc., with significantly fewer image-text pairs compared to SAM. Although vast attention has been paid to investigating the huge foundation model with vast data collection, given this impressive performance, we believe that WOVSS is a fascinating research topic that merits future investigation.

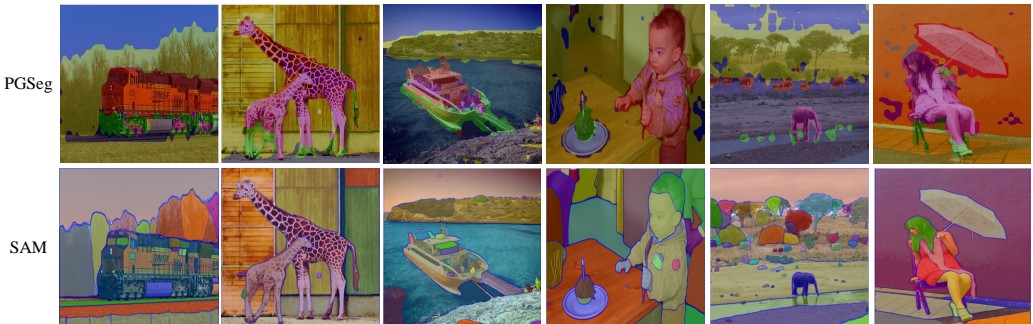

Figure 7: Class-agnostic segmentation comparison between SAM and PGSeg on LVIS.

## 7 Conclusion

The majority of efforts in *weakly open-vocabulary semantic segmentation* (WOVSS) have focused on implementing explicit grouping recognition while overlooking the customized supervision needed for the inherent clustering of group tokens. This oversight has led to a *granularity inconsistency* between the training and inference stages. To address this issue, our paper proposed to leverage *prototypical knowledge* to mine out explicit supervision, which encourages the generation of compact and rich feature representations for the group tokens. Additionally, we introduced multi-modal prototypes derived from image-text information to instantiate this knowledge, resulting in diverse patterns that enhance the segmenting ability of the group tokens. Through quantitative and qualitative experiments, we have demonstrated the effectiveness and superiority of this straightforward concept. In all, our bottom-up concept further validates the potential of prototypes, which exhibit *compactness* and *richness*, as promising elements for the top-down segmentation methods. Therefore, we believe that it is worth further exploring the full potential of prototypes in more weakly supervised tasks.

# 8 Acknowledgement

This work is supported by the National Key R&D Program of China (No. 2022ZD0160703), STCSM (No. 22511106101, No. 22511105700, No. 21DZ1100100), 111 plan (No. BP0719010) and National Natural Science Foundation of China (No. 62306178).

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
