# Supplementary Materials

# Contents

# 1  Computational Complexity of NPR

Recall that we claim the computational complexity of the prototype generation process in NPR is $\mathcal{O}(q \times m \times d)$, which is acceptable to implement. Furthermore, the experimental results in Section 5.3 also show that our proposed PGSeg indicates a reasonable level of computational efficiency. Here we aim to make a detailed deduction on this. The computational complexity of the prototype generation process is led by the EM process in Eq.(1) and (2), which could be divided into four stages:

1. The first stage could be regarded as a matrix multiplication operation between the prototypes $\boldsymbol{P} \in \mathbb{R}^{q \times d}$ and prototypical source feature $\boldsymbol{V} \in \mathbb{R}^{m \times d}$, and the computational complexity for that is $\mathcal{O}(q \times m \times d)$.

2. The second stage is the normalization process to obtain the estimated $\mathbf{Y}$, which could be treated as a SoftMax operation as presented in Eq.(1), and the complexity of such this operation is $\mathcal{O}(q \times m)$.

3. The third stage is to calculate the inner product between $y_{ij}$ and $v_j$, namely $\mathbf{Y}\boldsymbol{V}^{\top}$, resulting in the complexity of $\mathcal{O}(q \times m \times d)$. Based on an extra normalization operation, the final complexity reaches $\mathcal{O}(q \times m \times d) + \mathcal{O}(q \times m)$

4. The final step is to reconstruct the prototypical source by multiplying $\mathbf{Y}^{\top}$ with $\boldsymbol{P}$, which is specifically used in I-NPR, leading to the computational complexity of $\mathcal{O}(q \times m \times d)$.

Overall, the total computational complexity of the prototype generation process is the summation of the results above, i.e., $T(\mathcal{O}(q \times m \times d) + \mathcal{O}(q \times m) + \mathcal{O}(q \times m \times d) + \mathcal{O}(q \times m) + \mathcal{O}(q \times m \times d))$, where $T$ is the total iterations. Consequently, the estimated computational complexity is $\mathcal{O}(q \times m \times d)$ if neglecting the higher-order terms and constant factors, which concludes the proof.

# 2  Additional Experiments and Analysis

## 2.1  Datatset and Models

**Datasets.** In Section 5, we evaluate our PGSeg on five prevalent benchmarks, which are PASCAL VOC12 2012 [6], COCO [9], PASCAL Context [10], ImageNet-S [7], and LVIS [8]. Here is the detailed introduction of these five datasets as follows:

- **PASCAL VOC2012 [6]:** The PASCAL VOC12 dataset consists of a diverse collection of images spanning 21 different object categories (including one background class), such as a person, car, dog, and chair. The dataset provides annotations for both training and validation sets, with around 1,464 images in the training set and 1,449 images in the validation set. We use the validation set for the downstream evaluation. During the inference, we set the background score as 0.95.

- **COCO [9]:** The COCO Object dataset covers a wide range of 80 object categories, such as cars, bicycles, people, animals, and household items. For semantic segmentation, it has 118,287 training images and 5,000 images for validation. Correspondingly, we merely use the validation set, and the background score is 0.85.

- **Context [10]:** The dataset contains a diverse set of images taken from various scenes, including indoor and outdoor environments. It covers 59 common object classes, such as a person, car, bicycle, and tree, as well as 60 additional stuff classes, including sky, road, grass, and water. It has 118,287 training images and 5,000 images for validation. Here we merely consider the object dataset part, and use the validation set. The background score during the inference is set as 0.36.

- **ImageNet-S [7]:** ImageNet-S, distilled from the ImageNet [4], is a human-annotated pixel-level dataset specifically used for semantic segmentation. ImageNet-S has three versions based on the category amount, and we use the version with the maximum number of classes, which contains 919 classes and 12,419 validation samples. For a simple validation, we reduce the number of validation samples to 5,000. The background score is set to 0.11.

- **LVIS [8]:** The LVIS (Large Vocabulary Instance Segmentation) dataset is a large-scale dataset specifically constructed from COCO. It focuses on instance-level segmentation, where the goal is to identify and segment individual objects within an image. It includes a comprehensive vocabulary of over 1,200 object categories, making it one of the largest instance segmentation datasets available. It has 5,000 samples for evaluation. We ignore the instance-level annotation in the samples to proceed with semantic segmentation. The background score is set to 0.35.

**Models.** The PGSeg comprises an image encoder and a text encoder. The text encoder follows from [12] and consists of 12 transformer layers, each with a hidden dimension of 256. The text encoder adopts a *lower-cased byte pair encoding* (BPE) to encode the text with a vocabulary of 49,512 words. For the image encoder, we turn to the ViT-S with 12 transformer layers, each having a multi-head (6) self-attention and an MLP. We use layer normalization [1] to the input of each PG Unit. 3 Transformer layers are added to the final output of the PG Unit. In T-NPR, we select the text embedding before the mapping MLP, used to align with image embedding, as the prototypical source. In I-NPR, we use the input token, fed before the transformer layers, after a layer normalization as the image-level prototypical source. During the training stage, any pre-trained model is not used for both the image and text encoder. Several data augmentation approaches are employed, such as Random Flip and Color Normalization. During the inference, we directly adopt the model without any fine-tuning or training. Besides, we strictly follow [16, 13, 17] to set the input image size as $448 \times 448$, and adopt a stride strategy to generate the segmentation mask. The whole implementation is built on PyTorch [11] and MMSegmentation [3].

## 2.2 VOC results

Figure 1 presents more results of our PGSeg in VOC12. It is found that our PGSeg shows powerful grouping capability when segmenting the object-centric images. Besides, the learned group tokens could help segment objects in a compact and dense manner, which means there is less redundancy and noise in objects.

## 2.3 COCO results

Figure 2 presents some visualized results of COCO Object. Clearly, it has been observed that our PGSeg is able to perform fine-grained segmentation in the multi-object case. However, PGSeg is unable to completely capture some small objects, such as the bottle and plate in the image of the fourth row. This is essentially due to the wrongly-segmented group tokens, leading to some noise.

## 2.4 Context results

Figure 3 shows several visualized results of Context. Compared to COCO, the results of Context is comparably promising: the group tokens are distinctive from each other, capturing the whole object in a complete manner. Nevertheless, the semantic output seems to be not satisfying. On its face, such an issue is caused by a fix score of the background. Therefore, some areas in the group tokens with low scores, are directly recognized as the background. The dog in the third row could be an intuitive illustration. Besides, the over-segmentation phenomenon is quite severe in the multi-objects case, which is also a huge challenge in WOVSS.

## 2.5 LVIS & ImageNet-S results

Figure 4 shows the visualized results of both LVIS and ImagetNet-S. In ImageNet-S, we find that the group token in our PGSeg is powerful to finely cluster the object-centric object. However, due to the limited vocabulary during the training stage, our PGSeg is unable to match the input text with the corresponding semantic groups. In LVIS, it has been observed that over-segmentation and the presence of noisy regions are inherent issues that have emerged, similar to those observed in the Context dataset. Besides, the confidence scores of some complex group areas are still not high, leading to an object-level under-segmentation.

## 2.6 Two benefits

Here we incorporate more results on the analysis of two benefits, i.e., *compactness* and *richness*, to better understand NPR. Figure 5 illustrates the visualized t-SNE results of input patch tokens. The label IDs for these tokens are provided by the group token. Upon observation, it is evident that in comparison to GroupViT, the group tokens in PGSeg demonstrate a better ability to form a compact foundation. This aids in densely clustering the input patch tokens while being free from noise, thereby reducing redundancy. For *richness*, Figure 6 reports the dimensional distribution of the group token in level 2, which has 8 group tokens in total. Clearly, the dimensional variance of our PGSeg is larger than that of GroupViT, leading to a diverse feature representation.

To investigate the effects of each NPR strategy in terms of visual and textual parts, we have added additional analysis to examine the influence of I-NPR and T-NPR on compactness and richness in Figure 7 and 8. We find that both components contribute to compactness and richness, but As shown in Figure 7, I-NPR exerts a more pronounced impact on amplifying compactness. This observation aligns with the intuitive understanding that the image information embedded within I-NPR is inherently structured, thereby leading to more cohesive clustering. As shown in Figure 8, T-NPR has a more significant effect on improving richness, since it leads to a larger dimensional variance compared to I-NPR. We conjecture that the prototypes, which are sourced from text embeddings, could impose stronger semantic regularization on the group tokens. The results also underscore the complementary roles of I-NPR and T-NPR, further substantiating their importance in PGSeg.

## 3 Broader Impacts

Note that our training datasets, CC12M [2] and RedCaps12M [5], are sourced from the Internet. Consequently, the collection of these datasets raises concerns regarding privacy if not appropriately regulated. Additionally, text supervision typically relies on human annotators, which can introduce biases, intentional or unintentional, if the annotators are not impartial. It is key to address these issues through proper data regulation, privacy protection measures, and meticulous selection on the annotated information to ensure fairness and relieve potential biases.

## 4 Limitations

While our proposed PGSeg model can be applied to segment various downstream datasets, it shows relatively poor performance compared to methods that use additional supervision, such as segmenting masks. Particularly, PGSeg exhibits subpar performance in datasets like LVIS and ImageNet-S, indicating its limited capability for fine-grained segmentation in real-world scenarios. Moreover, our model is trained from scratch using training datasets that primarily consist of common object categories like dogs, people, etc. As a result, its applicability may be limited in specific domains such as medical [15] and LiDAR [14] images. Therefore, further investigation is warranted to assess the segmenting ability and potential application scope of our model in the future.

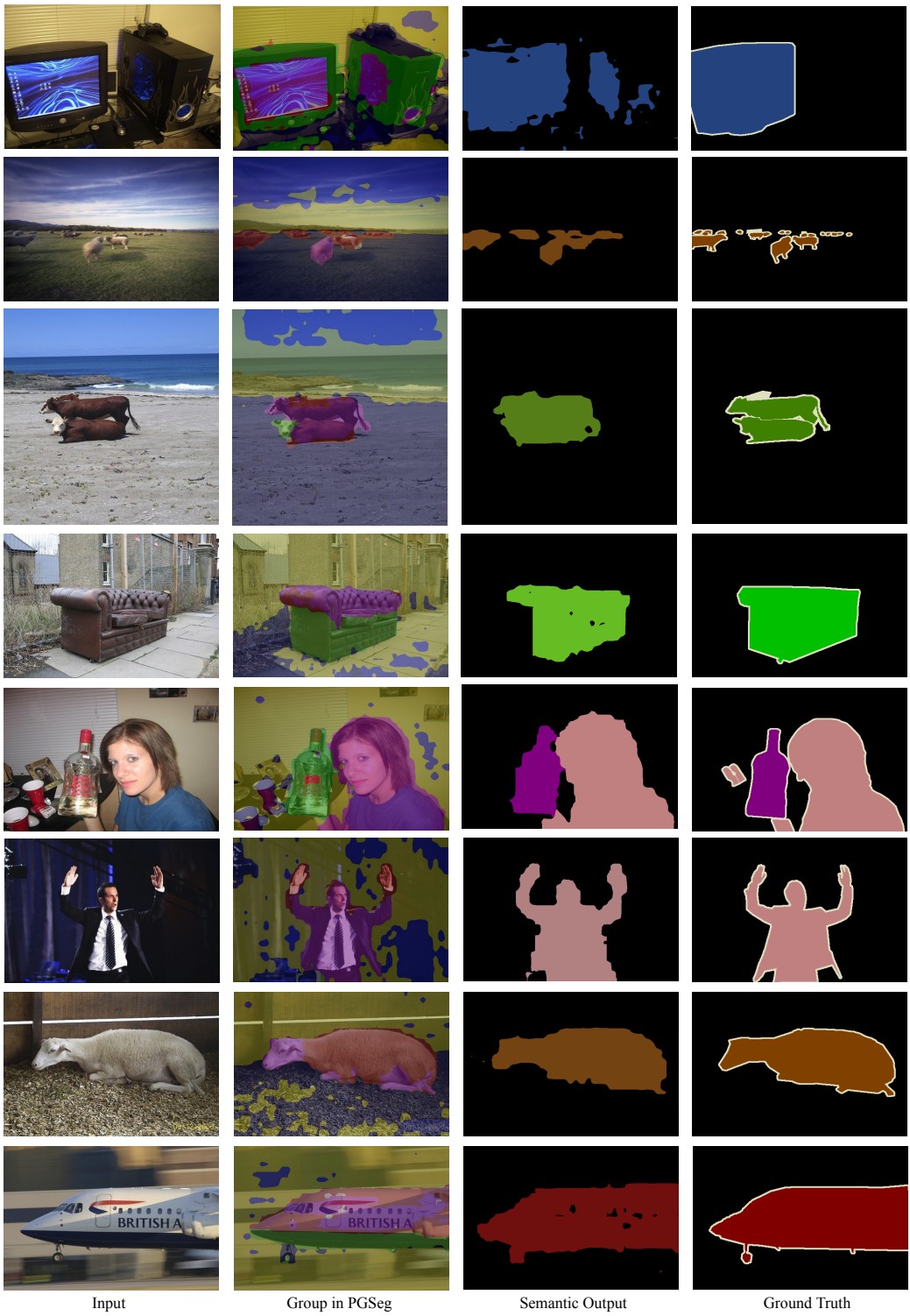

Input          Group in PGSeg          Semantic Output          Ground Truth

Figure 1: Qualitative results on PASCAL VOC12.

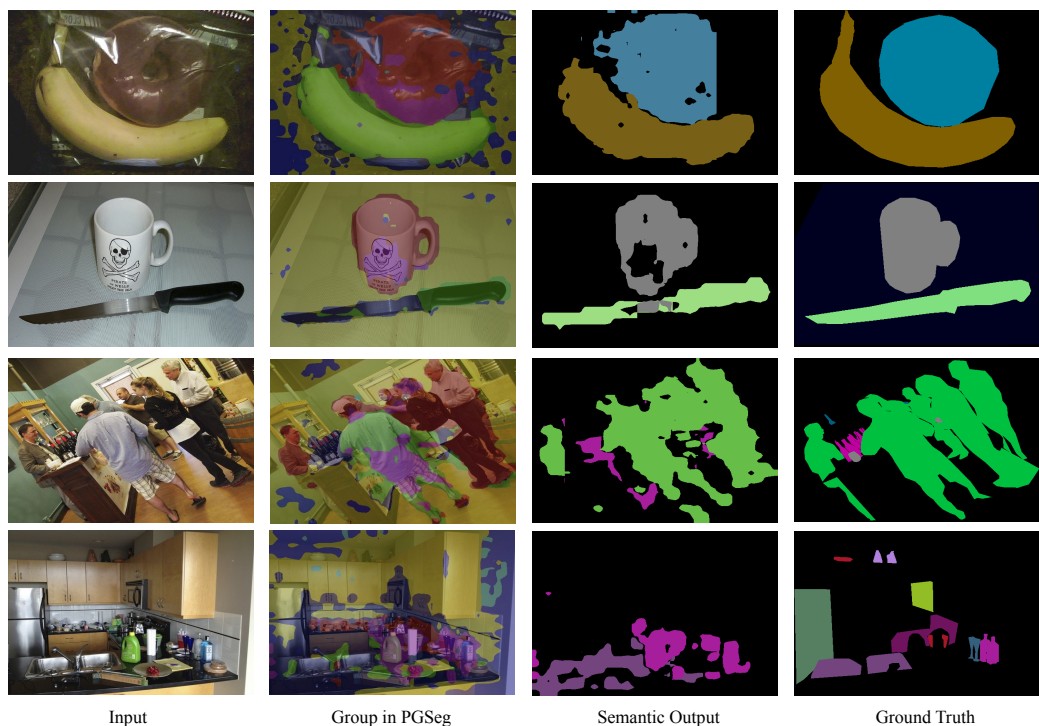

Input     Group in PGSeg     Semantic Output     Ground Truth

Figure 2: Qualitative results on COCO Object.

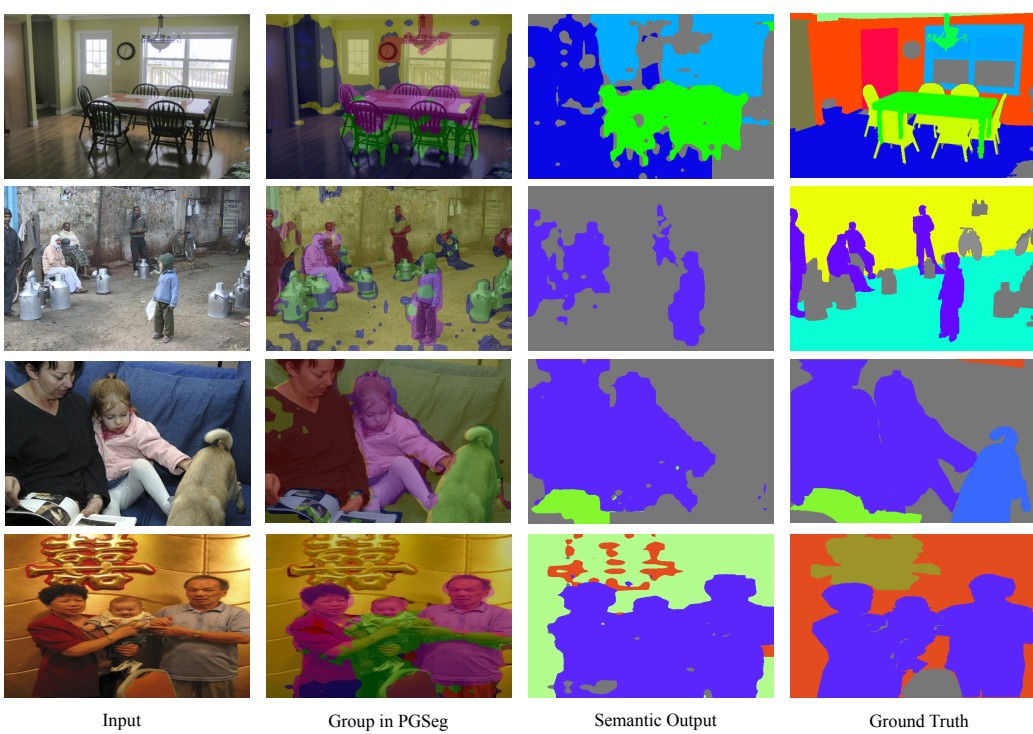

Input     Group in PGSeg     Semantic Output     Ground Truth

Figure 3: Qualitative results on Context.

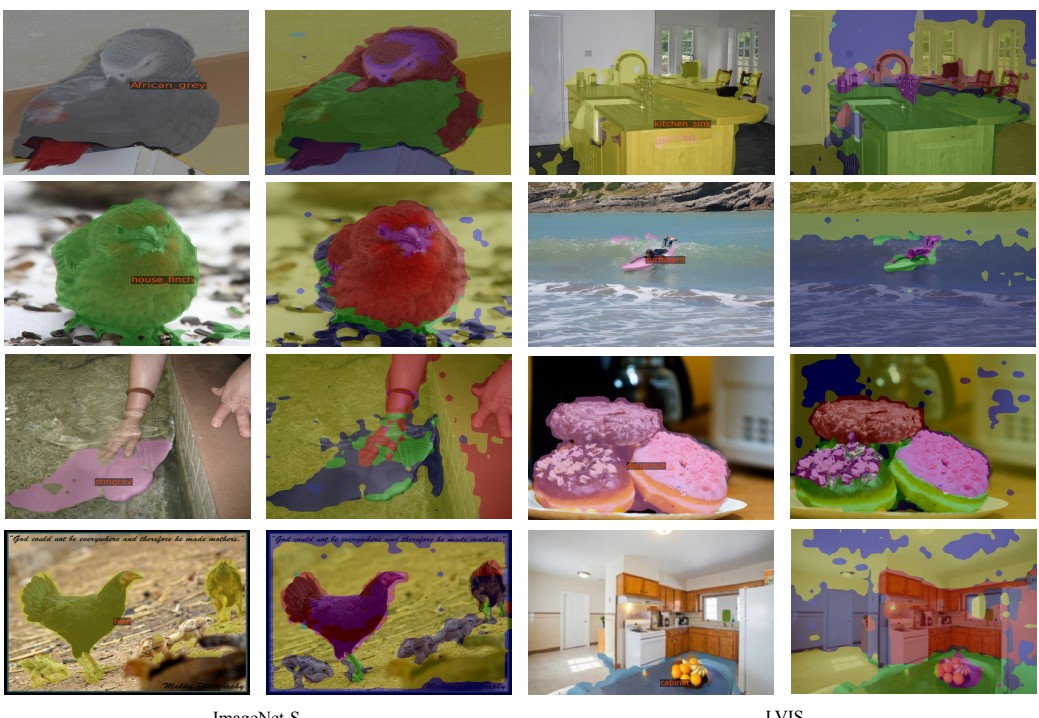

ImageNet-S                                              LVIS

Figure 4: Qualitative results on ImageNet-S and LVIS.

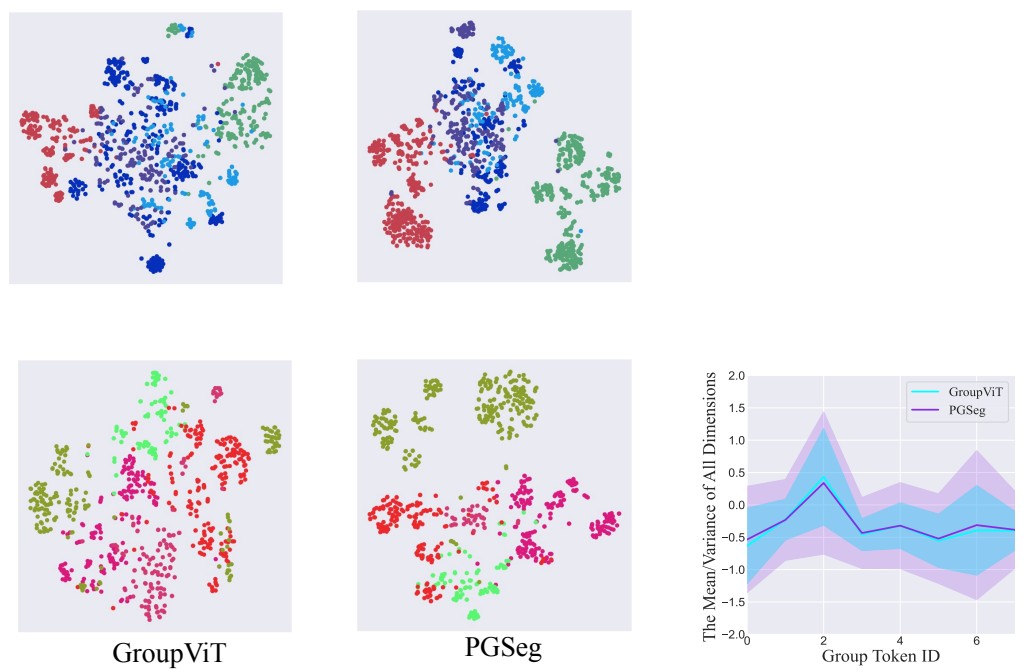

GroupViT                        PGSeg

Figure 5: *Compactness* analysis. The results come from 5 clustered patch tokens based on the group tokens.

Figure 6: Dimension distributions of each group token.