# OpenReview forum: "Uncovering Prototypical Knowledge for Weakly Open-Vocabulary Semantic Segmentation"
_NeurIPS.cc/2023/Conference — NeurIPS 2023 poster_

### Official Review · Reviewer_hVxT · 2023-07-05

**Soundness:** 3 good
**Presentation:** 2 fair
**Contribution:** 3 good
**Rating:** 5
**Confidence:** 4

**Summary:**

This paper proposes that the traditional WOVSS approach lacks effective supervision of group tokens. In this paper, visual patch tokens are modeled through the GMM model to obtain prototype features that can be used to supervise each group token. Experimental results show that the proposed PGSeg achieves state-of-the-art performance on several benchmark datasets.

**Strengths:**

1. Appling explicit supervision for each group token by constructing prototypes is novel.
2. The proposed PGSeg can reach good performance without large vision-language foundation models.


**Weaknesses:**

Major weakness:
1. As shown in Table 3, The performance of the baseline (GroupViT) drops a lot when using CC12M+RedCaps compared to CC12M+YFCC14M(Setting in the original GroupViT). To demonstrate the effectiveness of PGSeg, the performance of PGSeg trained with CC12M+YFCC14M should be reported.
2. Prototype P is an important parameter of this article. If prototype P is not updated during training, how much impact will it have on PGSeg performance?
3. The proposed PGSeg heavily relies on the EM algorithm to estimate the parameters of GMM, which may makes the model hard to train. What is the training time cost of PGSeg compared to the baseline(GroupViT).

Minor weakness:
1. Text in Figure 2 and Figure 3 are too small, which makes the workflow of PGSeg hard to read.


**Questions:**

See weaknesses.

**Limitations:**

No limitations are discussed.

---

> ### Author Rebuttal · Authors · 2023-08-09
>
> ### Q1
> >As shown in Table 3, The performance of the baseline (GroupViT) drops a lot when using CC12M+RedCaps compared to CC12M+YFCC14M(Setting in the original GroupViT). To demonstrate the effectiveness of PGSeg, the performance of PGSeg trained with CC12M+YFCC14M should be reported.
>
> Thanks for this comment! The requested results are reported in the following table:
> |Method | VOC12   |   Context     |  COCO  |
> |:----------:|:------:|:------:|:------:|
> |GroupViT (YFCC14M + CC12M)| 52.3 | 22.4 | 20.9 |
> |ViewCO (YFCC14M + CC12M)| 52.4 | 23.0 | 23.5|
> |PGSeg (YFCC14M + CC12M)| **53.9** | **23.5** | **24.7** |
>
> **(1)** As can be seen,  with the training data of (YFCC14M + CC12M), PGSeg consistently outperforms GroupViT and ViewCO across all three datasets. *It enables us to make the conclusion about the superiority of PGSeg more rigorously.*
>
> **(2)** Regarding the performance drop of GroupViT on COCO (27.5% -> 20.9%) when replacing YFCC14M with RedCaps, we observe a similar performance drop of PGSeg (28.7% -> 24.7%). We posit this might be attributed to the semantic descriptors of entities in RedCaps12M having a **more aligned resonance** with COCO labels compared to those in YFCC14M (please refer to the examples presented of **Fig. III** in the uploaded **rebuttal_figure.pdf**).
>
>
> ***
>
> ### Q2
> >Prototype P is an important parameter of this article. If prototype P is not updated during training, how much impact will it have on PGSeg performance?
>
> Thanks for highlighting this point. To address your concern, we conduct the experiments to verify PGSeg w/ or w/o updating prototypes and summarize the performance in the following table. According to the results, w/o updating, the model suffers from **a decrease of approximately 1.5% mIoU**, which confirms the pivotal role of the prototype updating mechanism.
> |Method | w/ update   |   w/o update |
> |:------------------:|:------:|:------:|
> |PGSeg (CC12M)| 53.2 | 51.8 (	$\downarrow$  **1.4**) |
> |PGSeg (RedCaps12M + CC12M)| 49.0 | 47.5 (	$\downarrow$ **1.5**)  |
>
> ***
>
> ### Q3
> >The proposed PGSeg heavily relies on the EM algorithm to estimate the parameters of GMM, which may makes the model hard to train. What is the training time cost of PGSeg compared to the baseline(GroupViT).
>
> Thanks for the comment. As discussed in **Section 4.1 (Complexity Analysis) and Table 6**, our EM algorithm is **not expensive.** With RedCaps12M+CC12M, the training time cost of GroupViT is **1.9 hours per epoch** and that of PGSeg is **2.5 hours per epoch**, which is at a reasonable level.
>
> ***
>
> ### Q4
> > (Minor weakness) Text in Figure 2 and Figure 3 are too small, which makes the workflow of PGSeg hard to read.
>
> Thanks for pointing this out. We will improve it.

---

> > ### Author Response · Authors · 2023-08-12
> > **A minor mistake in Q2**
> >
> > We are sorry for a mistake that the descriptions on the results reported in **Q2** are inadvertently reversed, where the modified version is as follows:
> >
> > |Method | w/ update   |   w/o update |
> > |:------------------:|:------:|:------:|
> > |PGSeg (RedCaps12M + CC12M) | 53.2 | 51.8 (	$\downarrow$  **1.4**) |
> > |PGSeg ( CC12M)| 49.0 | 47.5 (	$\downarrow$ **1.5**)  |
> >
> >
> > We sincerely regret this mistake, and apologize the inconvenience this may have caused.

---

### Official Review · Reviewer_YVCC · 2023-07-05

**Soundness:** 3 good
**Presentation:** 3 good
**Contribution:** 3 good
**Rating:** 6
**Confidence:** 5

**Summary:**


This paper presents a method for addressing the challenge of granularity inconsistency in weakly open-vocabulary semantic segmentation (WOVSS). To tackle this issue, the authors propose a technique called non-learnable prototypical regularization (NPR). NPR leverages explicit supervision from prototypical knowledge to guide group tokens, leading to improved object segmentation characterized by enhanced compactness and richness. Extensive experiments have been conducted to validate the effectiveness of this method on several benchmark datasets.


**Strengths:**

- The authors' use of prototypical knowledge to tackle the performance bottleneck caused by granularity inconsistency is a well-motivated and intriguing idea.

- The effectiveness of the proposed method is demonstrated through comprehensive experiments on multiple benchmark datasets. The results surpass those of previous works, establishing a new state-of-the-art performance.

**Weaknesses:**

- The term "granularity inconsistency" is initially unclear in the paper, specifically in lines 37-40 and the caption of Fig.2. Although Sec.3 provides clarification, it would be preferable to have a clear explanation in the abstract or introduction, as it serves as the main motivation for the proposed method. Alternatively, improving the caption of Figure 2 would help better understanding.

- In Sec.5, there is a lack of explanations for the experimental results. For instance, in the "Comparison with Zero-shot Baselines" section, there are insufficient explanations regarding why specific methods (e.g., ViT-8S) were chosen as baselines and the implications of the results beyond superior performance. To provide a more comprehensive analysis, the experiments section should include additional explanations elucidating the intended message conveyed by the results.

- Conducting ablation studies to evaluate the benefits of each NPR strategy (T-NPR only or I-NPR only) in terms of compactness and richness would be valuable. This would offer a clearer understanding of the effectiveness of each module by confirming the respective impacts of the NPR strategies on compactness and richness.

- (minor) In Fig.~4 (bottom), the airplane and contrail exhibit similar colors (purple and red), making it difficult to ascertain whether PGSeg is superior to GroupViT. To demonstrate the superiority of the proposed method, it would be helpful to differentiate the airplane and contrail more distinctly by modifying their colors or exploring alternative visual representation approaches.






**Questions:**

Please see the weakness above.

**Limitations:**

No, the authors did not discuss the limitations and potential negative social impact in the main text.

---

> ### Author Rebuttal · Authors · 2023-08-09
>
> ### Q1
> >The term "granularity inconsistency" is initially unclear in the paper, specifically in lines 37-40 and the caption of Fig.2. Although Sec.3 provides clarification, it would be preferable to have a clear explanation in the abstract or introduction, as it serves as the main motivation for the proposed method. Alternatively, improving the caption of Figure 2 would help better understanding.
>
> Thank you for this helpful suggestion! We will improve the caption of Fig. 2 as *"Illustration of granularity inconsistency in SGM. During the training phase, group tokens follow an all-to-one manner to be averaged to align with text embedding. However, during the inference stage, each group token is directly used to match the class embedding, i.e., the one-to-one alignment."* Meanwhile, the Introduction part will be improved as well to further clarify this aspect.
>
> ***
>
> ### Q2
> >In Sec.5, there is a lack of explanations for the experimental results. For instance, in the "Comparison with Zero-shot Baselines" section, there are insufficient explanations regarding why specific methods (e.g., ViT-8S) were chosen as baselines and the implications of the results beyond superior performance. To provide a more comprehensive analysis, the experiments section should include additional explanations elucidating the intended message conveyed by the results.
>
> Sorry for the confusion. Here we provide a detailed analysis of Table 1, which shall be incorporated in the updated version.
>
> 1. **Motivations for baseline design**. First, for **ViT-S based baselines** (rows 1-4 in Table 1), they are widely adopted in recent works, *e.g.*, [ref2, ref3], based on a simple idea: train a ViT and a text encoder jointly with an image-text contrastive loss (as CLIP [ref1]), and subsequently perform non-parametric grouping of final-layer pixel features using, *e.g.*, k-means, to obtain segmentation.
>
>     Second, for **ViT-8S** (row 5), it is inspired by findings in [ref4]: increased [CLS] tokens in ViT serve as meaningful centers for weakly supervised segmentation. We then increase the [CLS] token number of ViT-S to 8 to create a more potent parametric baseline ViT-8S (than ViT-S) to examine our approach.
>
>
>
> 2. **More result analysis.** Despite effectiveness, ViT-8S is limited in lacking explicit grouping recognition/regularization. SGM-based solutions relieve this issue and show consistently better performance against ViT-8S, which confirms the importance of integrating a grouping component into ViT for WOVSS. By further addressing granularity inconsistency in SGM, our PGSeg leads to substantial performance improvements against existing methods, *i.e.*, **5.8\%** and **3.3\%** over GroupViT and ViewCo.
>
>
> ***
>
>
> ### Q3
> >Conducting ablation studies to evaluate the benefits of each NPR strategy (T-NPR only or I-NPR only) in terms of compactness and richness would be valuable. This would offer a clearer understanding of the effectiveness of each module by confirming the respective impacts of the NPR strategies on compactness and richness.
>
> Thanks for the suggestion! We have added additional analysis to examine the influence of I-NPR and T-NPR on compactness and richness. Please refer to the enclosed **rebuttal_figure.pdf** for visualized plots.
>
> We find that both components contribute to compactness and richness, but:
> 1. As shown in **Fig. I**, **I-NPR exerts a more pronounced impact on amplifying compactness.** This observation aligns with the intuitive understanding that the image information embedded within I-NPR is inherently structured, thereby leading to more cohesive clustering.
>
> 2. As shown in **Fig. II**, **T-NPR has a more significant effect on improving richness**, since it leads to a larger dimensional variance compared to I-NPR. We conjecture that the prototypes, which are sourced from text embeddings, could impose stronger semantic regularization on the group tokens.
>
> The results also underscore the complementary roles of I-NPR and T-NPR, further substantiating their importance in PGSeg. Relevant discussions will be added to the paper.
>
> ***
>
> ### Q4
> >(minor) In Fig.~4 (bottom), the airplane and contrail exhibit similar colors (purple and red), making it difficult to ascertain whether PGSeg is superior to GroupViT. To demonstrate the superiority of the proposed method, it would be helpful to differentiate the airplane and contrail more distinctly by modifying their colors or exploring alternative visual representation approaches.
>
> Thanks! We agree and will improve it in the updated version.
>
> ***
>
> ### Reference
> >[ref1] Learning Transferable Visual Models From Natural Language Supervision, ICML2021.
>
> >[ref2] Viewco: Discovering text-supervised segmentation masks via multi-view semantic consistency, CVPR2022.
>
> >[ref3] GroupViT: Semantic Segmentation Emerges from Text Supervision, CVPR2022.
>
> >[ref4] Multi-class Token Transformer for Weakly Supervised Semantic Segmentation, CVPR2022.

---

> > ### Comment · Reviewer_YVCC · 2023-08-18
> >
> > Thanks for the detailed responses regarding our review.
> > Providing the rationale for choosing the ViT-8S baseline and the impact of the NPR strategy would strengthen your paper.
> >
> > After reading other reviewers' comments, I am confident of keeping my original ratings, weak accept.

---

> > > ### Author Response · Authors · 2023-08-18
> > > **Response to Reviewer YVCC**
> > >
> > > Thanks for your response,  and we will incorporate the suggested analysis  of ViT8-S and NPR strategy in the updated version.

---

### Official Review · Reviewer_nyAz · 2023-07-06

**Soundness:** 3 good
**Presentation:** 3 good
**Contribution:** 3 good
**Rating:** 5
**Confidence:** 3

**Summary:**

This paper studies the problem of weakly open-vocabulary semantic segmentation, which aims to segment objects of arbitrary classes using only image-text pairs. The paper proposes a prototypical guidance segmentation network, which uses non-learnable prototypes estimated from image and text features to regularize the group tokens that cluster the image patches. The paper shows that the proposed method improves the segmentation results with increased compactness and richness of the group tokens, and achieves state-of-the-art performance on several benchmark datasets.

**Strengths:**

1. The paper introduces a novel concept of non-learnable prototypical regularization. The idea of the paper is reasonable and novel.


2. The motivation and the methods are illustrated clearly in Introduction.


3. The performances improvement on VOC and COCO are significant.


**Weaknesses:**

1. Ablation studies are not enough. The number of prototypes should be more for ablation study. The affect of the  position and number of PG-Unit should be explored in ablation studies.

2. In Table 3, PGSeg shows the superiority on two datasets  VOC and COCO, but on Context, the performance of PGSeg is lower than pervious methods.


3. Fig.3 is not clear enough to illustrate the proposed method.  The text in the picture is too small to read.

**Questions:**

1. The comparison is not fair enough in Table 3 since the training data are different. Adding PGSeg(CC12M+YFCC14M) is fair to compare the proposed method and ViewCO and GroupViT.

2. Why the proposed method performs much better on  VOC and COCO but much lower on Context? What is the uniqueness of this dataset?

3. Some previous methods that tackles the semantic segmentation task using GMM should be discussed. e.g., "GMMSeg: Gaussian Mixture Models for Deep Generative Semantic Segmentation"

**Limitations:**

The authors did not addressed the limitations. I suggest to add more experiments and ablation studies.

---

> ### Author Rebuttal · Authors · 2023-08-09
>
> ### Q1
> >Ablation studies are not enough. The number of prototypes should be more for ablation study. The affect of the position and number of PG-Unit should be explored in ablation studies.
>
> Thanks for this suggestion! Here we first present the improved Table 5 by expanding the range of the number of prototypes. The settings in the following table are identical to Table 5 (please refer to L318-L321, page 8 for more details). Note that here the **group token number remains fixed** (64/8 for the 1st/2nd PG Unit). The result shows that **the optimal prototype number shall be set equally to the group token number.**
> |Prototype Number (1st\2nd Level)| 4 | **8** |16|32|64|
> |:----------:|:--------------:|:----------:|:--------------:|:----------:|:----------:|
> | 16 | 51.78 | 52.1 | 51.42 | - | -
> | 32 | 52.12 | 52.49 | 51.94 | 51.33 | -
> | **64** | 52.71 | **52.83** | 52.23 | 51.87 | 51.56|
> | 128 | 52.21 | 52.62 | 52.14 | 51.64 | 51.33|
> | 256 | 52.04 | 52.33 | 51.97 | 51.37 | 51.14|
>
> Secondly, we report the effect of the position of PG Unit on PGSeg with CC12M. Based on the following results, it is found that the **optimal position for the 1st (2nd) PG Unit in ViT-S is the 6th (9th) layer**, which is also consistent with the finding from GroupViT and ViewCO.
> |PG Unit Position (The layer position of the 1st/2nd PG Unit at ViT-S)|  Performance on VOC12 (%)|
> |:----------:|:--------------:|
> | (5/8) | 47.3 |
> | (6/9) | **49.0** |
> | (7/10) | 47.9 |
> | (4/11) | 48.3 |
>
> Finally, we present the ablation studies on the number of PG Unit in ViT as follows (Here the number of group tokens in the 11th layer is 4). As shown in the following table, the presence of **more or fewer PG Units would lead to some performance loss**, indicating the importance of the PG Unit amount in PGSeg.
> |PG Unit Number--(PG Unit position)| Performance on VOC12 (%)|
> |:----------:|:--------------:|
> | 1--(6) | 	45.2 |
> | 2--(6, 9) | **49.0** |
> | 3--(6, 9, 11) | 47.2 |
>
>
> We will add all these analyses in the updated version.
>
> ***
>
> ### Q2
> >In Table 3, PGSeg shows the superiority on two datasets VOC and COCO, but on Context, the performance of PGSeg is lower than pervious methods.
>
> Thanks for the comment, but there seems to be a misunderstanding. While it is true that PGSeg exhibits lower performance against MaskCLIP and SegCLIP on Context, it is crucial to consider the **substantial difference in training pipelines:** MaskCLIP and SegCLIP are both built on a powerful foundation model, *i.e.*, **CLIP**, which contains rich knowledge pattern from **~400M** image-text pairs. In contrast, our proposed PGSeg is **trained from scratch with simply 12M/24M** image-text pairs. Despite the significant data volume gap, PGSeg achieves **comparable performance on Context** and even **outperforms them on VOC12/COCO.** This actually underscores our strength of **data-efficient learning** rather than the weakness of our approach. Notably, when evaluating the methods under the same level of training data volume, PGSeg is the best across all three datasets.
>
> ***
>
> ### Q3
> >Why the proposed method performs much better on VOC and COCO but much lower on Context? What is the uniqueness of this dataset?
>
> Compared to VOC and COCO, where a majority of their foreground classes represent distinct entities (like dog, cat, etc.), Context additionally includes some foreground classes, such as "building" and "wall", which can easily be misconstrued as the "background" (Please refer to the **Supplementary Material** for some samples and their annotations of Context). Such distinct essence requires **superior recognition among these (near-)"background" classes** to achieve better performance on Context. The following table validates that PGSeg is unable to well segment these "background"-like classes, correspondingly leaving the room for our proposed PGSeg to segment on complex scenes.
> |"background"-like classes in Context |  Wall | Building |ceiling |ground |Average
> |:----:|:-------:|:--------:|:----------:|:----------:|:----------:
> | PGSeg (CC12M, 20.8)| 0.4 |5.1 |4.4|7.3|*4.3*
>
> ***
>
> ### Q4
> >The comparison is not fair enough in Table 3 since the training data are different. Adding PGSeg(CC12M+YFCC14M) is fair to compare the proposed method and ViewCO and GroupViT.
>
> Thanks for the comment. The requested results are reported in the following table. It is evident that PGSeg consistently surpasses GroupViT and ViewCO under the (CC12M+YFCC14M) setting. *Based on the results, we can more confidently conclude the superior performance of PGSeg.*
>
> |Method | VOC12   |   Context     |  COCO  |
> |:----------:|:------:|:------:|:------:|
> |GroupViT (YFCC14M + CC12M)| 52.3 | 22.4 | 20.9 |
> |ViewCO (YFCC14M + CC12M)| 52.4 | 23.0 | 23.5|
> |PGSeg (YFCC14M + CC12M)| **53.9** | **23.5** | **24.7** |
>
>
> ***
>
> ### Q4 & Q5
> >Fig.3 is not clear enough to illustrate the proposed method. The text in the picture is too small to read.
>
> >Some previous methods that tackles the semantic segmentation task using GMM should be discussed. e.g., "GMMSeg: Gaussian Mixture Models for Deep Generative Semantic Segmentation".
>
>
> Thanks for pointing these out. We will improve the figure and add the missing citations in the updated version.
>
> ***
> ### Reference
>
> >[ref1] Unsupervised domain adaptation for semantic segmentation via class-balanced self-training, CVPR2018.
>
> >[ref2] A Closer Look at Self-training for Zero-Label Semantic Segmentation, CVPR2021.
>
> >[ref3] Learning from future: A novel self-training framework for semantic segmentation, Neurips2022.

---

### Official Review · Reviewer_ae73 · 2023-07-12

**Soundness:** 3 good
**Presentation:** 2 fair
**Contribution:** 3 good
**Rating:** 5
**Confidence:** 3

**Summary:**

The paper addresses the problem of weakly open-vocabulary semantic segmentation, which is to segment images of novel classes without any segmentation annotation.
To segment unseen classes, a model is trained to align between image features and text caption to learn to associate between image patches and class names.
The proposed work argues that there is an discrepancy between visual-textual alignment during training and pixel-wise grouping during inference.
Thus, the paper proposes a non-learnable prototypical regularization (NPR), which leverages Gaussian Mixture Model to provide pixel-wise grouping supervision to better align the model for segmentation task.
Experiments are conducted on PASCAL, COCO, ImageNet and LVIS datasets.

**Strengths:**

+ The idea of direction of weakly open-vocabulary semantic segmentation is impactful since it can significantly reduce the annotation effort especially for the costly segmentation tasks.
+ The proposed direction of using Gaussian Mixture Model to group image patches into groups for additional training supervision is sensible and interesting especially this doesn't require any supervision.
+ The paper is self-contained and provides enough details for result reproduction.

**Weaknesses:**

+ The reviewer finds that the emphasis on the non-learnable aspect of the algorithm is misleading. Specifically, the proposed method does use expectation-maximization to compute the prototype which is a form of learning algorithm, which contradicts with the claim about non-learnable prototype regularization. Moreover, the reviewer is not convinced on why "non-learnable" is a desirable feature for segmentation given given that learnable/optimizable components would make the model fully differentiable and can be easily trained with gradient based optimization. Can the paper clarify this?
+ The experimental results appears to be inconclusive. Although the paper shows that the proposed method can achieves strong improvement with limited number of training data (using only CC12M table 3), but when scaling to large dataset, the improvement diminishes to less than 1%. Can the paper provide justification on this?
+ It is quite surprising that the model seems to be insensitive to the number of groups used during training (as shown in Table 5) where the difference is less than 1%? The number of groups should strongly influence performances as too many groups could introduce training noise and too few would not capture the full information in image. Can the paper justify this?

**Questions:**

Please refer to the weakness section

**Limitations:**

Sufficiently addressed

---

> ### Author Rebuttal · Authors · 2023-08-09
>
> ### Q1
> >The reviewer finds that the emphasis on the non-learnable aspect of the algorithm is misleading. Specifically, the proposed method does use expectation-maximization to compute the prototype which is a form of learning algorithm, which contradicts with the claim about non-learnable prototype regularization. Moreover, the reviewer is not convinced on why "non-learnable" is a desirable feature for segmentation given given that learnable/optimizable components would make the model fully differentiable and can be easily trained with gradient based optimization. Can the paper clarify this?
>
> **(1)** Sorry for the confusion. Here we call prototypes ``non-learnable`` because they are **detached from the gradient updating** of the network and online updated in a momentum manner (please refer to L235-239, page 6). The concept and definition are recently used in [ref1] and we adhere to it. We will include this in the updated version for clarity.
>
> **(2)** For Reviewer's proposition on using ``learnable prototypes`` for regularization, while the method appears logical on the surface, it would **not perform as well as anticipated.** Note that one major motivation of our method is to regularize group tokens using prototypical knowledge of image/text. However, once learning prototypes and group tokens together, the **model can be easily degraded** (e.g., simply making prototypes and group tokens identical and loss will be zero), and consequently **no proper regularization will be imposed.** To verify this, we further design a new baseline in which learnable prototypes are introduced by **replacing the I-NPR and T-NPR with another two SGMs.** In this way, the groups in the newly-added SGM serve as the prototypes, which are then used to regularize the original group tokens. Results are shown in the following table, and demonstrate that the baseline, with this learnable prototypical regularization, **detracts significantly from the segmentation performance.**
> |Baseline (GroupViT)|  PGSeg w/ learnable prototypes | PGSeg |
> |:----------:|:--------------:|:--------------:
> | 41.1 (CC12M) | 35.5 (	**$\downarrow$ 5.6**)|**49.0** (**$\uparrow$ 7.9**)
> | 50.8 (CC12M+RedCaps12M) | 46.3 (**$\downarrow$ 6.5**) | **53.2** (**$\uparrow$ 2.4**)
>
> ***
>
>
>
>
>
>
>
> ### Q2
> >The experimental results appears to be inconclusive. Although the paper shows that the proposed method can achieves strong improvement with limited number of training data (using only CC12M table 3), but when scaling to large dataset, the improvement diminishes to less than 1%. Can the paper provide justification on this?
>
> Thanks for the comment. First of all, we'd like to highlight that **PGSeg in general scales well, i.e., on VOC12 and COCO, under the same data volume.** Here we show the performance of PGSeg with YFCC14M for a clear illustration.
>
> |Method | VOC12   |   Context     |  COCO  |
> |:----------:|:------:|:------:|:------:|
> |GroupViT (YFCC14M + CC12M)| 52.3 | 22.4 | 20.9 |
> |ViewCO (YFCC14M + CC12M)| 52.4 | 23.0 | 23.5|
> |PGSeg (YFCC14M + CC12M)| **53.9** | **23.5** | **24.7** |
>
> Therefore, the reduced improvement gap (less than 1%) pertains only to the Context dataset. Compared to VOC and COCO, where a majority of their foreground classes represent distinct entities (like dog, cat, etc.), Context additionally includes some foreground classes, such as "building" and "wall", which can **easily be misconstrued as the "background".** (Please refer to the **Supplementary Material** for some samples and their annotations of Context). Such distinct essence requires superior recognition among these (near-)"background" classes to achieve better performance on Context. Thus, this semantic uniqueness of Context may **trouble the prototypes to perform expected segmentation on these "background" classes**. To demonstrate that, we present the performance of PGSeg on some "background"-like classes in Context. As shown in the following table,  PGSeg could not distinctively segment these "background" classes, even with scaling up to RedCaps12M. This leaves room for potential improvement of our approach. We will add these analyses!
> |"background"-like classes in Context|  Wall | Building |ceiling |ground |Average
> |:----:|:-------:|:--------:|:----------:|:----------:|:----------:
> | PGSeg (CC12M, 20.6)| 0.4 | 5.1 | 4.4| 7.3|*4.3*
> | PGSeg (Redcaps12M+CC12M, 23.8)| 0.8 |4.7 |8.8|5.7|*5*
>
> ***
>
>
>
>
> ### Q3
> >It is quite surprising that the model seems to be insensitive to the number of groups used during training (as shown in Table 5) where the difference is less than 1%? The number of groups should strongly influence performances as too many groups could introduce training noise and too few would not capture the full information in image. Can the paper justify this?
>
> **(1)** Sorry for the misleading. We clarify that Table 5 is intended to validate the impact of **prototype number rather than group number**. As mentioned in the caption, **the numbers of group tokens remain fixed at 64 (8) for the 1st (2nd) PG Unit**. The conclusion from Table 5 is just that PGSeg is insensitive to **prototype number**. We will revise Table 5 to avoid any potential misunderstandings.
>
> **(2)** We totally agree with the Reviewer's insight that the number of group tokens is important to segmentation performance. As such, we have conducted an ablation study on the number of group tokens in PGSeg using CC12M. Note that we maintain the number of prototypes equal to the number of group tokens. It is evident in the following table that **too many or too few group tokens have detrimental effects on the segmentation performance.** We will add this analysis.
>
> |Group Token Number (at 1st----2nd PG Unit)|  Performance on VOC12 (%)|
> |:----------:|:--------------:|
> | 16----4 | 29.3 |
> | 16----8 | 39.5 |
> | 64----8 | **49.0** |
> | 128----8 | 43.3 |
> | 128----4 | 37.3 |
>
>
> ***
>
> ### Reference
>
>
> >[ref1] Rethinking Semantic Segmentation: A Prototype View, CVPR2022.

---

> > ### Comment · Reviewer_ae73 · 2023-08-21
> >
> > Dear authors,
> >
> > Thanks for clarifying my concerns in the rebuttal. As such, I will keep my original score.

---

> > > ### Author Response · Authors · 2023-08-21
> > > **Response to Reviewer ae73**
> > >
> > > Thanks for your response, and we will incorporate these analyses in the updated version.

---

### Author Rebuttal · Authors · 2023-08-09

## Summary



We thank reviewers for their valuable feedback, and appreciate the great efforts made by all reviewers, ACs, SACs and PCs.

We are invigorated by the **positive evaluation** from all reviewers. Specifically, they find the method **novel and reasonable** (nyAz, ae73), **well-motivated and intriguing** (YVCC, hVxT), the experimental results being **state-of-the-art** (YVCC) and showing  **significant improvements** (nYAz), the writing **self-contained** (ae73) and **effectively illustrated** (nYAz).


In response to the comments and suggestions, we have provided a detailed respective rebuttal for each reviewer, and here we summarize major points for convenience.

- We have extended ablation studies to investigate the effect of **group token number**(ae73),  **prototype number and PG-Unit position/number** (nyAz), **PGSeg with YFCC14M** (nyAz, hVxT), **prototype updating** (hVxT), as well as **the respective effect of T-NPR and I-NPR** (YVCC).
- We have conducted a more in-depth and comprehensive analysis of experimental design and results, including **indistinctive performance on Context** (ae73, nyAz), **training costs** (hVxT), and **baseline design** (YVCC).


All these will be merged into the article.

>Below is the uploaded **"rebuttal_figure.pdf"**

---

### Author Response · Authors · 2023-08-18
**Follow-up Discussion**

Dear all reviewers,

We sincerely appreciate all reviewers' constructive comments on our submission, which actually helps us improve our submission. As Discussion Stage 2 is about to end, we highly appreciate knowing if our responses have addressed your concerns. If you have any questions, comments, or concerns left, please let us know, we would like to provide more discussion with the reviewers. Thank you very much again.

Best,

The authors of Submission

---

> ### Comment · Area_Chair_CGV9 · 2023-08-18
>
> thanks, all responses have been read and will be taken into account

---

### Decision · Program_Chairs · 2023-09-21

**Decision:**

Accept (poster)

**Comment:**

All reviews are positive (1 weak accept, 3 borderline accept). The most confident reviewer is most positive. Reviewers praise the problem setting, key idea in the method, results, and presentation. The reviewers request clarifications and more comparisons, and two reviewers respond to explicitly state they maintain their positive rating after reading the rebuttal. For the two that did not respond, their concerns seem well addressed.